# An ideal rapid-cycle Thouless pump

**Savvas Malikis⋆ and Vadim Cheianov**

Instituut-Lorentz, Universiteit Leiden, Leiden, The Netherlands

⋆ malikis@lorentz.leidenuniv.nl

## Abstract

Thouless pumping is a fundamental instance of quantized transport, which is topologically protected. Although its theoretical importance, the adiabaticity condition is an obstacle for further practical applications. Here, focusing on the Rice-Mele model, we provide a family of finite-frequency examples that ensure both the absence of excitations and the perfect quantization of the pumped charge at the end of each cycle. This family, which contains an adiabatic protocol as a limiting case, is obtained through a mapping onto the zero curvature representation of the Euclidean sinh-Gordon equation.

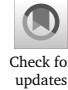
# 1  Introduction

Thouless pumping [1] serves as a fundamental example of topological quantization of transport in a many-body system. It occurs in a one-dimensional band insulator whose parameters are varied at an infinitesimally slow rate in such a way as to describe a non-contractible loop around the critical manifold of gapless states in the system's parameter space. If these conditions are met then an integer number of charge quanta per cycle is pumped through any given cross-section of the system. Thouless pumping is intimately related to the integer Hall effect [2], since in both cases the quantized transport coefficient is expressed in terms of the Chern index of a $U(1)$ principal bundle [3] associated with the Berry or Zak phase [4–6]. Also there is an extension to Floquet-Thouless energy pumps with the corresponding topological invariant [7]. Despite its conceptual importance, the Thouless pump had remained a hypothetical device until only recently when it was realized in highly controlled systems of ultracold bosons [8] and fermions [9].

The exact quantization of the pumped charge in a Thouless pump requires perfectly adiabatic driving which ensures that no elementary excitations are created during the pump cycle. An actual experiment is always performed at a finite driving frequency, which, in general, lifts the topological protection of charge quantization [10–12]. This is one of the reasons why the original Thouless pump is unlikely to supersede the quantum Hall and Josephon effects as the current standard [13–16]. Non-adiabatic effects on quantized transport in parametric pumps [17–21], of which the Thouless pump is a special case, were studied in [10–12, 22–26]. Generally, such effects consist in frequency-dependent corrections to the average pumped charge as well as non-equilibrium noise. Also worth mentioning are the finite-size corrections [27] and the trade-off between the requirements of adiabaticity and large system size [28].

Recently, attention has started to turn to ways of mitigating the non-adiabatic effects in finite-frequency protocols. Among the proposed strategies are the dissipation assisted pumping [29], non-Hermitean Floquet engineering [30, 31] and counterdiabatic control in both quantum [32, 33] and classical [34] settings. Also it is worth mentioning the non-adiabatic quantization of the current, instead of the charge, in quasiperiodic Thouless pumps [35]. Moreover, the topological classification of periodically driven systems [36, 37] has led to the proposal of novel topological phases; among others a 2D anomalous Floquet-Anderson insulator was proposed, that can serve as a 2D non-adiabatic charge pump [38].

In this work we take a complementary route and look into the optimization of the driving path in the pump's parameter space. Focusing on the paradigmatic model of the Thouless pump, the driven Rice-Mele insulator, we explicitly construct an infinite family of driving protocols which achieve both the exact quantization of the pumped charge and vanishing non-equilibrium noise at driving frequencies ranging from zero to the typical band gap of the insulator.

This paper is organized as follows: In Sec. 2 we introduce the Thouless pumping and highlight the main aspects of this work. In Sec. 3 we introduce the mathematical framework for quantum pumping. In Sec.4 we perform the mapping from the zero curvature representation to the Rice-Mele model. In Sec. 5 by using this mapping, we find the conditions for the existence of an ideal pump.

# 2 Overview of Thouless pumping and statement of the main result

## 2.1 Adiabatic and non-adiabatic Thouless pumping

We begin our discussion with a brief recapitulation of Thouless pumping in a one-dimensional insulator described by the paradigmatic Rice-Mele model [39].

The system consists of an 1-dimensional half-filled bipartite lattice described by the tight-binding Hamiltonian

$$\hat{H}_{\mathbf{p}} = \sum_{j=1}^{N} \Big[ m(a_j^\dagger a_j - b_j^\dagger b_j) + (t_1 a_j^\dagger b_j + t_2 b_j^\dagger a_{j+1} + h.c.) \Big]. \tag{1}$$

Here $\mathbf{p} = (m, t_1, t_2)$ is a point in the space of tight-binding parameters, where $m$ is the real staggering onsite potential and the alternating hopping amplitudes $t_{1,2}$ are generally complex. The operators $a, b, a^\dagger, b^\dagger$ are the canonical Fermion fields $\{a_i, a_j^\dagger\} = \{b_i, b_j^\dagger\} = \delta_{ij},$. Quantised pumping is expected to occur in an infinite system, however for technical purposes it is convenient to keep $N$ finite assuming periodic boundary conditions $j + N \equiv j$ and take the $N \to \infty$ limit when necessary. At any given point $\mathbf{p}$ in the parameter space, the single-particle energy spectrum of the Hamiltonian consists of two bands, which in semiconductor physics are called the valence band and the conduction band, see Fig 3(b) for some examples. Due to the half-filing condition, that is the overall number of fermions being $N$, the ground state of the system corresponds to a completely filled valence band and an empty conduction band. The ground state is separated from the excited states by the band gap $E_g$, which vanishes when $m = 0$ and $\delta = 0$, where $\delta = |t_1| - |t_2|$. By a slight abuse of notation we denote the critical $E_g = 0$ point in the $(m, \delta)$ subspace of the parameter space as $\mathbf{p}_c$.

Assume now that system is prepared in the ground state of the Hamiltonian $H_{\mathbf{p}}$ and then the parameters of the Hamiltonian are allowed to change very slowly as a function of time $\tau$ such that $\mathbf{p}(\tau)$ describes a closed loop in the parameter space, *i.e.* $\mathbf{p}(0) = \mathbf{p}(T)$. If the path $\mathbf{p}(\tau)$ avoids the critical point $\mathbf{p}_c$ at all times, then the band gap will ensure that the evolution of the system is adiabatic that is no elementary excitations are created in the process. Thus, due to the periodicity of $\mathbf{p}(\tau)$ the final state of the system will coincide with the initial one. Remarkably, notwithstanding the fact that no charge carriers are present in the system at any stage of the cycle, the protocol may result in charge transport across the system. [1] Furthermore, as was shown by Thouless [1] the charge pumped through any cross-section of the system in a given cycle is an integer coinciding with the winding number of the closed path around the critical point $\mathbf{p}_c$.

Adiabaticity of the pumping cycle is crucial for the exact topological quantization of pumped charge because any elementary excisions created in the process may skew the very low (one particle per cycle) average current in a non-universal way, also contaminating the experiment with non-equilibrium noise. In a laboratory experiment one aims to work at driving frequencies $T^{-1} \ll E_g$ [10] because the interband Landau-Zener transitions creating particle-hole pairs in the bulk of the pump should be suppressed in such a case. However, in large pumps, where the adiabatic quantization of the pumped charge is accurate, the necessary conditions for adiabaticity turn out to be more stringent [28]. Even though topologically quantised Thouless pumping does not generally survive away from the adiabatic limit [11], this does not in principle exclude the existence of fine-tuned noise-free quantized charge pumping protocols operating at finite frequency. For such a rapid-cycle protocol to exist, a number of conditions

---

[1]In a periodic chain, the charge transfer means that there is circulation of exactly one unit of charge per period, resulting in the same "snapshot" at $\tau = T$. In an open chain, it is assumed that there are charge reservoirs at the boundaries and the transport occurs through the chain.

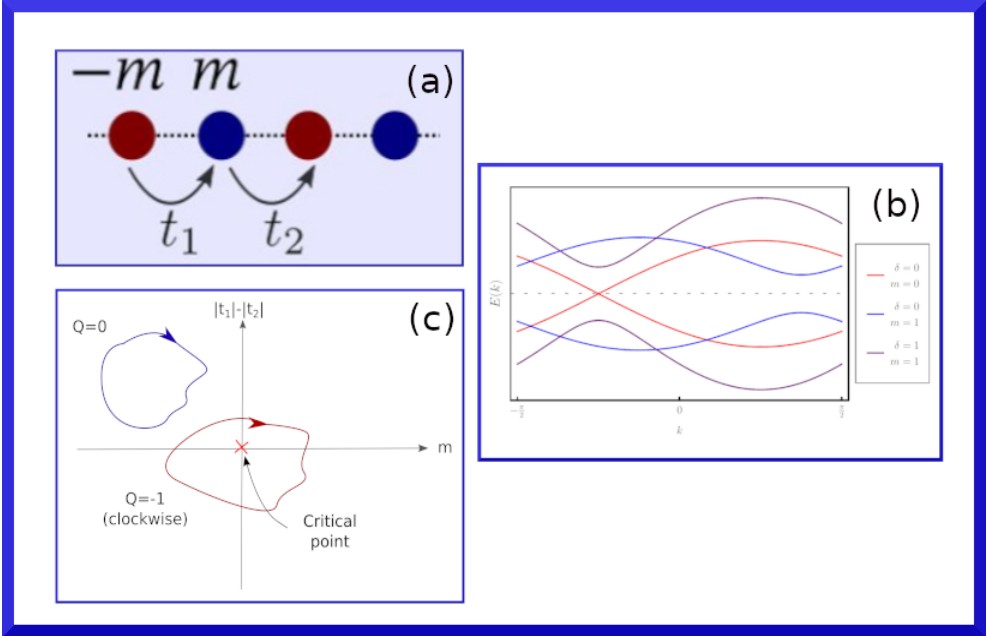

Figure 1: Panel (a): the Rice-Mele lattice. Panel (b): the energy bands for Rice-Mele Hamiltonian for different values of $m, \delta = |t_1| - |t_2|$ in the 1st Brillouin Zone. The critical point is found at $m = 0$ and $\delta = 0$. Panel (c): the pumped charge Q after the completion of a full adiabatic cycle is the winding number around the critical point in the parameter space of the Rice-Mele model.

need to be met. Firstly, for the reasons given above, it is necessary that at the end of the cycle the system should return to its initial state $|\Psi_0\rangle$, which is to say that

$$\hat{U}_T |\Psi_0\rangle = \mathcal{T} \exp\left(-i \int_0^T \hat{H}_{\mathbf{p}(\tau')} d\tau'\right) |\Psi_0\rangle = \exp(iG) |\Psi_0\rangle , \tag{2}$$

where $\hat{U}_T$ is the unitary operator of evolution over one cycle, often called the Floquet evolution operator, $\mathcal{T} \exp$ is the time ordered exponential, and $G$ is some real phase. In other words, the initial state $|\Psi_0\rangle$ needs to be an eigenstate of the Floquet operator $\hat{U}_T$. While Floquet eigenstates do undoubtedly exist, it is generally a hard task to initialise a many-body system in a Floquet eigenstate. This brings us to the second requirement, which is that $|\Psi_0\rangle$ has to be an easily initialisable state, preferably the ground state of a local Hamiltonian. Finally, even if a protocol is found such that $|\Psi_0\rangle$ is simultaneously an eigenstate of the Floquet evolution operator and is easy to initialise, there is no guarantee that the charge pumped in one cycle will not be zero. As is discussed in [11], the requirement of non-vanishing pumped charge is equivalent to a singular band crossing condition for the Bloch spectrum of the Floquet operator $-i \ln \hat{U}_T$ [11]. Due to the complex relationship between the path $\mathbf{p}(\tau)$ and the Floquet operator (2), it is generally unclear how to chose the trajectory $\mathbf{p}(\tau)$ in a way that would ensure the required band crossing or whether such a choice exists at all.

## 2.2 The proposed solution

In this work we present an explicit construction of an infinite family of closed paths $\mathbf{p}(\tau)$ in the parameter space of the Rice-Male Hamiltonian which obey all the above requirements for the rapid-cycle quantised Thouless pumping. For each element of the family, the pump is initialised in the ground state of the Rice-Mele Hamiltonian at some point $\mathbf{p}(0)$ of the parameter

space and then a closed loop $\mathbf{p}(\tau)$ is performed in a finite period $T$ such that $\mathbf{p}(T) = \mathbf{p}(0)$ and no excitations are created at the end of the cycle. A way that our work can be seen is that out of all possible cycles $\mathbf{p}(t)$, we find explicit examples that survive without requiring the adiabaticity protection. In the following paragraphs of this section we give a complete self-contained description of the family as well as illustrating the operation of the rapid-cycle protocol with a specific example. The rest of the article is devoted to a detailed mathematical derivation of our result.

A rapid pumping cycle is generated by the Rice-Mele hamiltonian (1) with the time-dependent parameters given in the following explicit functional form

$$m(\tau) = \frac{\dot{v}(\tau)}{2\sqrt{\mu-1}}\mathrm{dc}(ru(\tau),\mu), \tag{3}$$

$$t_{1,2}(\tau) = \frac{V(\tau)e^{-i\theta(\tau)}}{4}\Big[\mathrm{nc}(ru(\tau),\mu) \pm \mathrm{sc}(ru(\tau),\mu)\Big]. \tag{4}$$

In this expression $\tau$ is the time. The cycle begins at $\tau = 0$ and ends at $\tau = T$ at which point the tight binding parameters return to their initial values. The symbols $\mathrm{dc}, \mathrm{nc}$ and $\mathrm{sc}$ stand for the three minor Jacobi elliptic functions [40] having period $r$, which is related to the elliptic parameter $\mu > 1$ by $r = 4\sqrt{1/\mu}K(1/\mu)$, where $K$ is the complete elliptic integral of the first kind. The $+, -$ sign in the expression for $t_{1,2}$ corresponds to $t_1, t_2$, respectively. The functions $u$ and $v$ are arbitrary smooth functions of their time argument satisfying the boundary conditions

$$u(T) = u(0) + n, \qquad \dot{u}(T) = \dot{u}(0) = 0, \tag{5}$$

where $n$ is a nonzero integer and

$$v(0) = 0, \qquad v(T) = T. \tag{6}$$

The function $V$ is defined as

$$V(\tau) = \sqrt{[\dot{v}(\tau)]^2 + [\lambda\dot{u}(\tau)]^2}, \tag{7}$$

and the phase $\theta(\tau)$ is given by

$$\theta(\tau) = \arg[\lambda\dot{u}(\tau) + i\dot{v}(\tau)], \tag{8}$$

where $\lambda = r\sqrt{\mu-1}$.

In the following sections we give a detailed proof that if at $\tau = 0$ the system is prepared in the ground state of the Rice-Mele Hamiltonian and then the parameters of the Hamiltonian evolve according to Eqs. (3), (4) then at the end of the period $\tau = T$ the system will be found in the same ground state. Furthermore, we show that in the $N \to \infty$ limit the net charge pumped through any cross section of the system during one cycle will be given by the integer $n$, Eq. (5). For these reasons the protocol achieves noisless quantised pumping at a finite frequency. The protocol is obviously fine-tuned because an arbitrary finite frequency cycle would generate excitations in the system. However, the family of available loops $\mathbf{p}(\tau)$ is parametrised by two arbitrary (up to the boundary conditions) functions $u$ and $v$ and two real parameters $\mu$ and $T$ and therefore is very large. The $T \to \infty$ limit connects this family with the family of adiabatic Thouless protocols. We finally note, that although no excitations are found in the system at $\tau = T$, intermittent excitations are created during the cycle. Our peculiar choice of the path $\mathbf{p}(\tau)$ ensures complete annihilation of all particle-hole pairs at the end of the cycle without complete time reversal of the evolution as the latter would have resulted in vanishing pumped charge.

We note that the ideal pump can operate at frequencies comparable to the typical value of the band gap, although we do not establish the rigorous upper bound on the frequency in the present work. We also note that contrary to the Thouless protocol using real $t_{1,2}$. the rapid-cycle protocol requires complex hopping amplitudes with time-dependent arguments. Physically, this corresponds to the application of a uniform electric field pulse $E = \dot{\theta}(\tau)$ or, in the context of ultracold atomic pumps, an acceleration pulse along the chain.

Lastly, one reason why the proposed rapid-cycle protocol functions away from the adiabatic limit is that the ground state of the Rice-Mele Hamiltonian at $\tau = 0$ is also an Eigenstate of the Floquet operator $\hat{U}_T$; by definition, the Floquet eigenbasis satisfies the necessary condition (2). It is known that if this condition is satisfied then the finite-frequency corrections to the pumped charge are non-analytic in frequency, $\delta q = o(\omega^\infty)$, [11] that is they vanish faster than any power law when frequency approaches zero. Thus, finding a natural way to prepare the many-body system in a Floquet eigenstate would already constitute a substantial improvement in the performance of a finite frequency pump. However, our protocol brings it one step further also protecting the crossing of the Bloch-Floquet bands, which removes the non-analytic in frequency corrections too making the quantization of the pumped charge perfect.

Next, we illustrate our general result with an example.

## 2.3 Example

We consider the protocol defined by Eqs. (3)-(8) with

$$v(\tau) = \tau, \qquad u(\tau) = w(\alpha\tau), \tag{9}$$

where $\alpha$ is the inverse period $\alpha = 1/T$ and we have defined the function

$$w(z) = z - (2\pi)^{-1} \sin(2\pi z). \tag{10}$$

One can easily see that the boundary conditions (5) and (6) are satisfied with $n = 1$. By straightforward substitution one finds the function $V$, Eq. (7),

$$V(\tau) = \sqrt{1 + 4\lambda^2\alpha^2 \sin^4(\pi\alpha\tau)},$$

and the shape of the electric field pulse

$$E(\tau) = \dot{\theta}(\tau) = \frac{2\pi\alpha^2\lambda \sin(2\pi\alpha\tau)}{4\alpha^2\lambda^2 \sin^4(\pi\alpha\tau) + 1}. \tag{11}$$

We recall that $\lambda$ in these expressions is a free real parameter, which also determines the elliptic parameter $\mu$ and the parameter $r$ in Eqs. (3) and (4). They now take the form

$$m(\tau) = \frac{1}{2\sqrt{\mu - 1}} \text{dc}(rw(\alpha\tau), \mu), \tag{12}$$

$$t_{1,2}(\tau) = \frac{V(\tau)e^{-i\theta(\tau)}}{4} [\text{nc}(rw(\alpha\tau), \mu) \pm \text{sc}(rw(\alpha\tau), \mu)],$$

where the parameter $r$ is related to the elliptic parameter by $r = 4\sqrt{1/\mu}K(1/\mu)$ and $\lambda = r\sqrt{\mu - 1}$.

We now proceed to the results of the numerical simulation illustrating the operation of the protocol for a particular choice of the remaining free parameters $\alpha$ and $\mu$. The results of the simulation for $\alpha = 0.09$ and $\mu = 3.5$ in a Rice-Mele chain of $N = 50$ sites are shown in Fig. 2. Plotted are the time-dependent tight-binding parameters, pumped charge and the adiabatic fidelity, which is the magnitude of the projection of the time-dependent state $|\Psi(\tau)\rangle$ onto the

instantaneous vacuum state of the time-dependent Hamiltonian $H_{\mathbf{p}(\tau)}$. To illustrate the fine-tuned nature of the ideal protocol, the same data are shown for a non-ideal protocol with the same $|t_1|$, $|t_2|$ and $m$ but with $\theta = 0$. One can see that in contrast to the non-ideal case, at the end of each cycle of the ideal protocol exactly one unit of charge is pumped across the system and wave function returns to the Hamiltonian's exact vacuum state. Curiously enough, the proposed protocol does not seem to have any finite size corrections at least to the numerical accuracy of our computation.

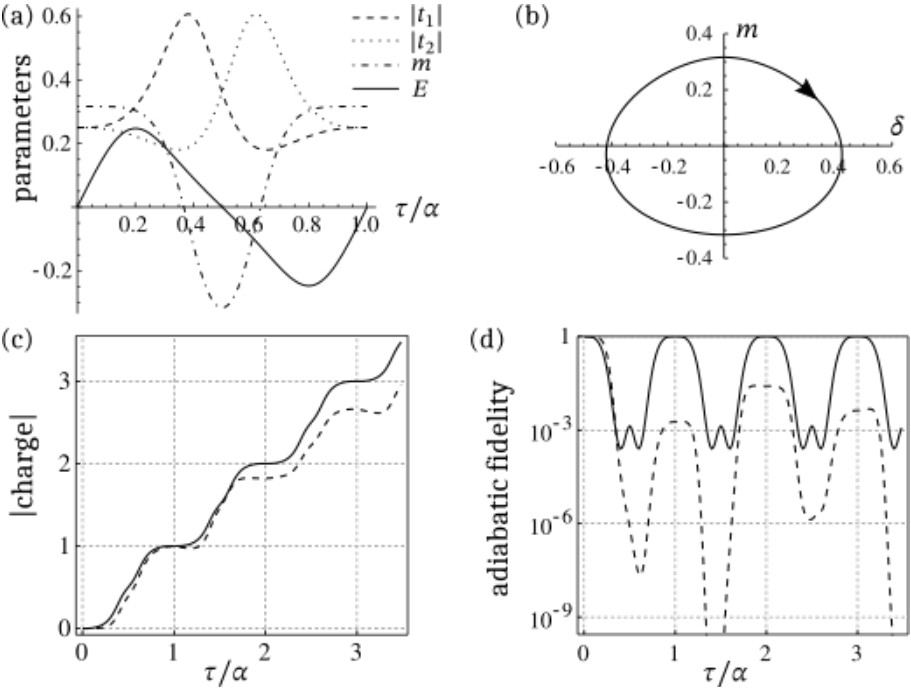

Figure 2: Comparison of an ideal and a non-ideal pumping protocols for the driving parameters $\alpha = 0.09$ and $\mu = 3.5$, and the system size $N = 50$. The non-ideal protocol is obtained from the ideal one by suppressing the electric field pulse. Panel (a): Time-dependent tight-binding parameters and the electric field $E = \dot{\theta}$ for the ideal protocol. Panel (b): Projection of the driving path onto the $(\delta, m)$ plane. Panel (c): Pumped charge as a function of time for the ideal (solid line) and non-ideal (dashed line) protocols. Panel (d): Adiabatic fidelity for the ideal (solid line) and non-ideal (dashed line) protocols.

The rest of the paper deals with the rigorous definition and the proof of the remarkable properties of the ideal protocol discussed in this section and illustrated in Fig. 2.

## 3  Mathematical framework

The Hamiltonian (1) admits for complete separation of variables in the Fourier space $\hat{H}_p = \sum_{k \in \mathcal{B}} [\hat{n}_+(k)\epsilon_+(k) + \hat{n}_-(k)\epsilon_-(k)]$, where $\mathcal{B} = (2\pi/N)\{0, 1, \ldots N-1\}$ is the discretised Brillouin zone, $\epsilon_\pm(k)$ are the quasiparticle energies $\epsilon_-(k) = -|\epsilon_+(k)|$, and $\hat{n}_\pm(k)$ are the quasiparticle occupation number operators. The vacuum state of the Hamiltonian Eq. (1) is a simultaneous eigenstate of all occupation numbers such that $n_-(k) = 1$ and $n_+(k) = 0$ for all $k \in \mathcal{B}$. Generally, the vacuum of the Hamiltonian (1) is separated from the lowest-energy excited state by the band gap $E_g = 2\sqrt{\delta^2 + m^2}$, where we used the notation $\delta = |t_1| - |t_2|$. The condition of non-vanishing of the band gap defines the non-simply connected parameter space

of the Rice-Mele insulator $\mathcal{P} = \{(m, t_1, t_2)|E_g > 0\}$.

A pumping protocol consists of initialization of the system in a state $|\Psi\rangle$ and a continuous loop $\mathbf{p} : [\tau_1, \tau_2] \to \mathcal{P}$, $\mathbf{p}(\tau) = (m, t_1, t_2)(\tau)$, which defines a time-dependent Hamiltonian $H(\tau) = H_{\mathbf{p}(\tau)}$. Thus the initial state evolves based on $|\Psi(\tau)\rangle = \hat{U}(\tau, \tau_1)|\Psi\rangle$ where $\hat{U}(\tau, \tau_1)$ is the unitary evolution operator generated by $H(\tau)$. Normally, indefinite periodic repetition of the protocol is implied so we shall refer to $\hat{U}(\tau_2, \tau_1)$ as the Floquet operator.

For any system that is prepared at its vacuum state, its evolution is constrained to the invariant eigenspace of the Hamiltonian (1) defined by the constraints $n_+(k) + n_-(k) = 1$, for all $k \in \mathcal{B}$. This subspace is isomorphic to $\bigotimes_{k \in \mathcal{B}} \mathcal{V}_k$, where $\mathcal{V}_k$ is the two-dimensional single-particle Hilbert space corresponding to the Bloch momentum $k$. Using this representation of $\mathcal{V}$, we can write a natural initial wave function as $|\Psi\rangle = \bigotimes_{k \in \mathcal{B}} |v(k)\rangle$, where $|v(k)\rangle$ is the negative-energy eigenvector of the initializing Hamiltonian in the single-particle subspace $\mathcal{V}_k$, and the end-of-the-cycle wave function as

$$|\Psi(\tau_2)\rangle = \hat{U}(\tau_2, \tau_1)|\Psi\rangle = \bigotimes_{k \in \mathcal{B}} \hat{U}(\tau_2, \tau_1, k)|v(k)\rangle \,. \tag{13}$$

We will omit the arguments $\tau_2, \tau_1$ for shortness from now on; the evolution will be assumed from an initial $\tau_1$ to a final $\tau_2$. We note that for any $N$ the functions $|v(k)\rangle$ and $\hat{U}(k)$ appearing on the right hand of (13) can be viewed as restrictions of smooth $N$-independent maps $|v\rangle : [0, 2\pi] \to \mathbb{C}^2$ and $\hat{U} : [0, 2\pi] \to \mathrm{SU}(2)$ to the discretized Brillouin zone $\mathcal{B}$. With this definition of $|v\rangle$ and $\hat{U}$ in mind, it has been shown [36] that for a protocol with the wave function $|\Psi(\tau_2)\rangle$ given in the form (13), given that it is an eigenvector of the Floquet operator $\hat{U}$, the amount of charge traversing any given cross-section of the system per cycle is given in the $N \to \infty$ limit by $\langle q \rangle = I[\hat{U}, |v\rangle\langle v|]$, where we have introduced the notation

$$I[\hat{U}, \hat{P}] = -i \oint \frac{dk}{2\pi} \mathrm{Tr}[\hat{P}(k)\hat{U}^{-1}(k)\partial_k \hat{U}(k)]. \tag{14}$$

If the state $|\Psi\rangle$ is eigenvector of $\hat{U}$, it means that $\hat{U}(k)|v(k)\rangle = \xi(k)|v(k)\rangle$, and $I[\hat{U}, |v\rangle\langle v|]$ coincides with the winding index of the continuous map $\xi : [0, 2\pi] \to \mathrm{U}(1)$. It is therefore an integer.

We shall call a pumping protocol *ideal* if the initial state is both the vacuum of $H(\tau_1)$ and eigenvector of $\hat{U}$ and, moreover, $I[\hat{U}, |v\rangle\langle v|] \neq 0$. Ideal protocols are easy to initialize, noise-free and they achieve integral quantization of the pumped charge. Today, the only known type of this pumping protocol is the adiabatic Thouless protocol [1], i.e. when the parameters $\mathbf{p} = \mathbf{p}(\alpha\tau)$ and $\alpha \to 0$. In that regime the pumping is known to be robust, i.e. insensitive in the particular choice of path. Now we are going to present another class of examples whose charge pumping survives away from the adiabatic limit.

## 4 Mapping Rice-Mele to zero curvature representation

The EsGE is a partial differential equation for a real field $\phi$

$$\Delta\phi + \sinh(\phi) = 0\,, \tag{15}$$

where $\Delta$ is the two-dimensional Laplace operator. It can be shown by direct calculation that EsGE is equivalent to the zero curvature condition

$$\partial_y \hat{A}_x - \partial_x \hat{A}_y + [\hat{A}_x, \hat{A}_y] = 0\,, \tag{16}$$

for the anti-Hermitian matrix-valued vector field

$$\hat{A}_x = \frac{1}{4}\left[\hat{\sigma}_+ \cosh\frac{\phi - ik}{2} - \hat{\sigma}_- \cosh\frac{\phi + ik}{2} + i\hat{\sigma}_3\,\partial_y\phi\right],\tag{17}$$

$$\hat{A}_y = \frac{i}{4}\left[\hat{\sigma}_+ \sinh\frac{\phi - ik}{2} + \hat{\sigma}_- \sinh\frac{\phi + ik}{2} - \hat{\sigma}_3\,\partial_x\phi\right],\tag{18}$$

where $\hat{\sigma}_i$ stands for the $i$th Pauli matrix, $\hat{\sigma}_\pm \equiv \hat{\sigma}_1 \pm i\hat{\sigma}_2$, and $k$ is the spectral parameter, which is allowed to take any complex value. The zero-curvature condition (16) implies, in particular, the existence of a globally defined unitary fundamental matrix $\hat{F}(\mathbf{x})$ which solves the overdetermined system of equations

$$\partial_x\hat{F} = \hat{A}_x\hat{F},\qquad \partial_y\hat{F} = \hat{A}_y\hat{F}\,.\tag{19}$$

Now, let $\gamma : [\tau_1, \tau_2] \to \mathbb{E}^2$, be a differentiable path in the Euclidean plane. Define the matrix:

$$\hat{h}_\gamma(\tau, k) = i\frac{d\gamma}{d\tau}\cdot\hat{A}(\mathbf{x}, k)|_{\mathbf{x}=\gamma(\tau)}\,,\tag{20}$$

and introduce a two-spinor field $\psi(k), \psi^\dagger(k), k \in \mathcal{B}$, satisfying the canonical anticommutation algebra $\{\psi_\alpha(k), \psi^\dagger_\beta(p)\} = \delta_{\alpha\beta}\delta_{kp}$ where $\alpha, \beta$ are spinor indices.

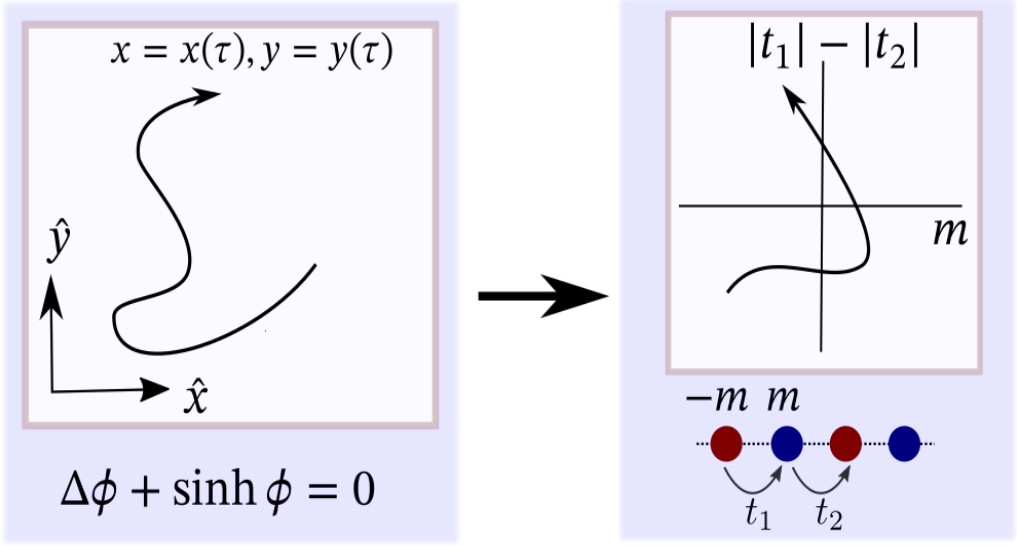

Figure 3: The mapping of the zero-curvature representation on Rice-Mele model. We specify the time-dependent parameters $t_i, m$, by choosing specific solution $\phi$, a path in x,y plane and using the relations (22) and (23)

Then the operator

$$H_\gamma(\tau) = \sum_{k\in\mathcal{B}}\psi^\dagger(k)\hat{h}_\gamma(\tau, k)\psi(k)\tag{21}$$

coincides with the reciprocal space representation of the Rice-Mele Hamiltonian Eq.(1) with the following choice of the tight-binding parameters

$$m(\tau) = -\frac{1}{4}\dot{\gamma}\times\nabla\phi(\gamma)\,,\tag{22}$$

where the cross stands for the skew-symmetric product, and

$$t_1(\tau) = \frac{\|\dot{\boldsymbol{\gamma}}\|}{4} e^{-\frac{\phi(\gamma)}{2} - i\theta}, \quad t_2(\tau) = \frac{\|\dot{\boldsymbol{\gamma}}\|}{4} e^{\frac{\phi(\gamma)}{2} - i\theta}, \tag{23}$$

where we used the polar decomposition of the velocity vector $\dot{\gamma}_x = \|\dot{\boldsymbol{\gamma}}\| \cos\theta$, $\dot{\gamma}_y = \|\dot{\boldsymbol{\gamma}}\| \sin\theta$. The transformation to the reciprocal space representation is given by the Fourier transform of the fields

$$\begin{pmatrix} a_j \\ b_j \end{pmatrix} = \frac{1}{\sqrt{N}} \sum_{k \in \mathcal{B}} e^{ikj} e^{-i(k+\pi)\frac{\hat{\sigma}_3}{4}} \psi(k).$$

Let $\phi(\mathbf{x})$ be some solution of equation (15) and consider a path $\boldsymbol{\gamma}$ starting at the point $\mathbf{x}_1 = \boldsymbol{\gamma}(\tau_1)$ and ending at $\mathbf{x}_2 = \boldsymbol{\gamma}(\tau_2)$. Define the unitary evolution matrix

$$\hat{U}_\gamma(k) = \hat{U}(\tau, \tau_1; k)\big|_{\tau = \tau_2}, \tag{24}$$

as the boundary value of the unitary evolution matrix solving the Schrödinger equation

$$i\frac{\partial \hat{U}_\gamma(\tau, \tau_1; k)}{\partial \tau} = \hat{h}_\gamma(\tau, k)\hat{U}_\gamma(\tau, \tau_1; k), \quad \hat{U}(\tau_1, \tau_1; k) = \hat{\mathbb{1}}. \tag{25}$$

It follows directly from equations (16), (19) and (20) that $\hat{U}_\gamma(k) = \hat{F}(\mathbf{x}_2)\hat{F}^{-1}(\mathbf{x}_1)$. In other words, the zero-curvature condition ensures that the unitary evolution operator $\hat{U}_\gamma(k)$ only depends on the initial and final points of the path [41]. So we managed to provide a map from the zero curvature representation of classical Sinh-Gordon field onto the time-dependent tight-binding quantum Hamiltonian (see Fig. (3)).

## 5 Setting the conditions for an ideal pump

We have given the mapping between the Rice-Mele model and the zero curvature representation. Till now, everything is pretty general. In order to specify a protocol, one needs a solution $\phi$ of (15) and a path on $\mathbb{E}^2$. In this section, we are going to find the conditions for Floquet-proper protocols. We now introduce the following definition

**Definition 1** *(Orderly path) Consider a two-dimensional Euclidean space endowed with a matrix-valued vector field $\hat{A}$ satisfying the zero-curvature condition (16). A differentiable path $\boldsymbol{\gamma} : [\tau_1, \tau_2] \to \mathbb{E}^2$ will be called* orderly *if $\forall k \in [0, 2\pi)$ (i) $\hat{h}_\gamma(\tau_1, k) = \hat{h}_\gamma(\tau_2, k) \neq 0$ and (ii) $[U_\gamma(k), \hat{h}_\gamma(\tau_{1(2)}, k)] = 0$* [2].

An orderly path ensures that the cycle fulfils the condition (2). For each orderly path $\boldsymbol{\gamma} : [\tau_1, \tau_2] \to \mathbb{E}^2$ we define its index

$$\nu_\gamma = I[\hat{U}_\gamma, \hat{P}_-], \tag{26}$$

where $\hat{P}_-(k)$ is the projector onto the negative-energy eigenspace of the Hamiltonian $\hat{h}_\gamma(\tau_1, k) = \hat{h}_\gamma(\tau_2, k)$.

We now proceed to considering a special class of solutions to the equation (15), namely the functions $\phi(\mathbf{x})$ which are translationally invariant along the x direction. Such functions obey the ordinary differential equation

$$\varphi_y'' + \sinh(\varphi) = 0. \tag{27}$$

---

[2]Any differentiable loop, that is a path satisfying $\boldsymbol{\gamma}(\tau_1) = \boldsymbol{\gamma}(\tau_2)$ and $\dot{\boldsymbol{\gamma}}(\tau_1) = \dot{\boldsymbol{\gamma}}(\tau_2)$, has $U_\gamma(k) = \mathbb{I}$ and is therefore orderly. Time-dependent Rice-Mele Hamiltonians constructed from such loops provide an interesting family of systems exhibiting a perfect dynamic localization [42].

Equation (27) coincides with Newton's second law for a particle with coordinate $\varphi$ moving in the potential $\cosh(\varphi)$ therefore all its solutions are periodic functions of $y$. Moreover, all such periodic solutions map out closed loops in the phase space $(\varphi, \varphi'_y)$ winding counterclockwise around the point of stable equilibrium $(0,0)$. This observation justifies the following proposition.

**Proposition 1** *Let $\varphi(y)$ be a non-vanishing solution of Eq. (27) having the fundamental period $\lambda$ and consider a path $p : [0,1] \to \mathbb{R}^2$, defined by $p(\tau) = (\delta, m)(\tau)$ where*

$$\delta(\tau) = \frac{1}{2} \sinh \frac{\varphi(n\lambda\tau)}{2}, \quad m(\tau) = -\frac{1}{4}\varphi'(n\lambda\tau), \tag{28}$$

*and $n \in \mathbb{Z}$. Then $p$ avoids the origin $(0,0)$ and its winding index around the origin coincides with $-n$.*

Now, let $\phi(\mathbf{x}) = \varphi(y)$ and let $\lambda$ be be the fundamental period of $\varphi(y)$. Consider a family of straight-line paths $\boldsymbol{\gamma}^\alpha : [0,1] \to \mathbb{E}^2$ defined by

$$\boldsymbol{\gamma}^\alpha(\tau) = \boldsymbol{x}_1 + \left(\alpha^{-1}\tau, n\lambda\tau\right), \tag{29}$$

where $\boldsymbol{x}_1 \in \mathbb{E}^2$ is a fixed starting point, $n \in \mathbb{Z}$, and $\alpha > 0$ is a real parameter. Denote by $\hat{G}_-(k)$ the projecttor onto the negative energy eigenspace of the Hamiltonian $\hat{g}(k) = i\hat{A}_x(\boldsymbol{x}_1, k)$.

**Lemma 1** *For the family of matrix-valued functions of $k$ $\hat{U}^\alpha(k) \equiv \hat{U}_{\gamma^\alpha}(k)$ associated with the family of paths (29) there exists a differentiable $2\pi$-periodic real-valued function $\chi(k)$ such that the limit*

$$\hat{\omega}(k) = \lim_{\alpha \to 0} e^{i\alpha^{-1}\chi(k)\hat{g}(k)}\hat{U}^\alpha(k) \tag{30}$$

*exists and $\hat{\omega}(k)$ satisfies the following two properties*
*(i) $[\hat{\omega}(k), \hat{g}(k)] = 0$ and*
*(ii) $I\left[e^{i(a+\chi b)\hat{g}}\hat{\omega}, \hat{G}_-\right] = -n$ for all $a, b \in \mathbb{R}$.*

The detailed proof of this Lemma is given in the Supplementary material. Its intuitive meaning is as follows. Let for simplicity $\boldsymbol{x}_1 = 0$. At small $\alpha$, the evolution along the path (29) is generated by the Hamiltonian $i\alpha^{-1}A_x(n\lambda\tau\mathbf{e}_y, k)$, which up to a constant is the time-dependent Rice-Mele Hamiltonian whose time-dependent parameters $\delta(\tau) = |t_2|(\tau) - |t_1|(\tau)$ and $m(\tau)$, are given in equation (28). A large pre-factor in front of the Hamiltonian ensures adiabatic evolution, so the Hamiltonian commutes with the unitary evolution matrix at each moment of time $\tau$. Also, in this limit $\hat{G}_- = \hat{P}_-$. Although, the $\alpha \to 0$ limit of the evolution matrix does not exist due to the rapid oscillations arising from the dynamical phases, one can define the transfer matrix (30), where the dynamical phase factors have been stripped away. The transfer matrix bears the information about the topological Berry phase associated with the adiabatic path of the system in the parameter space. It follows from Proposition 1 that such an adiabatic path performs to a $n$-fold clockwise winding around the critical point $(\delta, m) = (0,0)$. It is well established that for adiabatic paths of such type $I[\hat{\omega}, \hat{P}_-] = -n$, see e.g. [43]. Moreover, it is easily shown that $I[\hat{\omega}, \hat{P}_-] = I[e^{i\hat{Q}}\hat{\omega}, \hat{P}_-]$ for any differentiable Hermitian matrix $\hat{Q}(k)$, which commutes with $\hat{\omega}(k)$ and whose eigenvalues are periodic functions of $k$. Next, we consider a particular type of path.

**Definition 2** *Denote by $\Gamma_\mu$, the set of all differentiable paths $\boldsymbol{\gamma} : [\tau_1, \tau_2] \to \mathbb{E}^2$ satisfying the following boundary conditions: (i) $\gamma_y(\tau_2) = \gamma_y(\tau_1) + \mu$, (ii) $\dot{\gamma}_y(\tau_1) = \dot{\gamma}_y(\tau_2) = 0$, (iii) $\dot{\gamma}_x(\tau_1) = \dot{\gamma}_x(\tau_2) \neq 0$.*

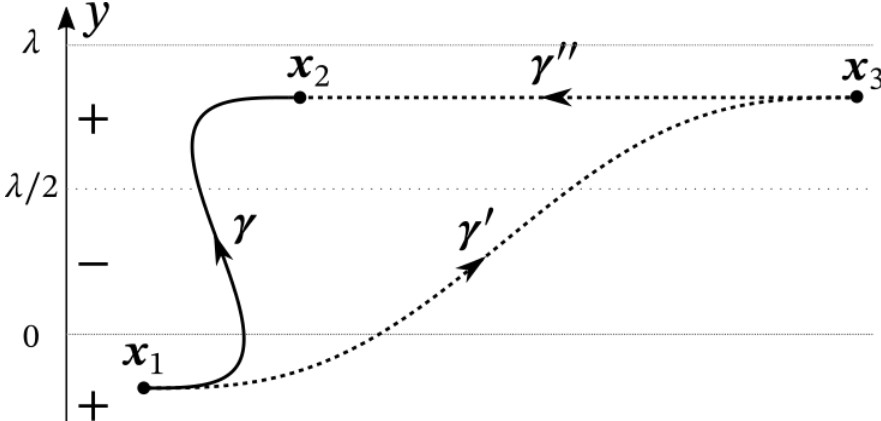

Figure 4: The two alternative paths from the starting point $x_1$ to the end point $x_2$. In the limit of large $\|x_3 - x_2\|$, evolution along $\gamma'$ is amenable to the asymptotic estimate (30), while subsequent evolution along $\gamma''$ has no effect on the winding index, Lemma 1 (ii).

**Theorem 1** *Let $\phi(\mathbf{x}) = \varphi(y)$ and let $\lambda$ be its fundamental period. Let $\gamma \in \Gamma_{n\lambda}$, where $n \in \mathbb{Z}$. Then (i) $\gamma$ is orderly, and (ii) $\nu_\gamma = -n$.*

**Proof of Theorem 1** *Let $x_{1(2)} = \gamma(\tau_{1(2)})$. First, we check that every element of $\Gamma_{n\lambda}$ is orderly. Items (ii) and (iii) of Definition 2 imply that $\hat{h}_\gamma(\tau_{1(2)}, k) = ic\hat{g}(k)$ where $c = \dot{\gamma}_x(\tau_{1(2)})$, and thanks to item (i) $\hat{g} = i\hat{A}_x(x_1, k) = i\hat{A}_x(x_2, k)$. It remains to verify condition (ii) of Definition 1. To this end, we pick a point $x_3 = x_2 + (1/\alpha, n\lambda)$ and construct an arbitrary differentiable path $\gamma'$ starting at $x_1$ and ending at $x_3$, and a straight-line path $\gamma''$ starting at the point $x_3$ and ending at $x_2$, see Fig. 4. Using the zero-curvature condition we can write $\hat{U}_\gamma(k) = \hat{U}_{\gamma''}(k)\hat{U}_{\gamma'}(k)$, where $\hat{U}_{\gamma''}(k) = e^{is\hat{g}(k)}$ and $s = \|x_2 - x_3\|$. Furthermore, by virtue of Lemma 1, $U_{\gamma'}(k) = e^{-i\alpha^{-1}\chi(k)\hat{g}(k)}\omega(k) + o(1)$.*

*It follows that $\hat{U}_\gamma(k) = e^{i[s-\alpha^{-1}\chi(k)]\hat{g}(k)}\omega(k) + o(1)$. Since the left hand side of this expression does not depend on $\alpha$, there exists a limit $\hat{S}(k) = \lim_{\alpha\to 0} e^{i[s-\alpha^{-1}\chi(k)]\hat{g}(k)}$ such that $U_\gamma(k) = \hat{S}(k)\omega(k)$. We can now write $\nu_\gamma = I[\hat{S}(k)\hat{\omega}(k), \hat{P}_-]$ as*

$$\nu_\gamma = \lim_{\alpha\to 0} I[e^{i[s-\alpha^{-1}\chi(k)]\hat{g}(k)}\hat{\omega}(k), \hat{P}_-] = I[\hat{\omega}(k), \hat{G}_-] = -n,$$

*where we have used item (ii) of Lemma 1 and the fact that $\hat{P}_- = \hat{G}_-$*

Theorem 1 yields the following Corollary, which constitutes the main result of this work.
**Corollary.** Let $\phi(\mathbf{x}) = \varphi(y)$, where $\varphi(y)$ is a solution to the equation (27) having the fundamental period $\lambda$, let $\gamma \in \Gamma_{n\lambda}$, where $n \in \mathbb{Z}$, and let $H(\tau)$ be a time-dependent Rice-Mele Hamiltoniain with the parameters given by Eqs. (22), (23). Then the pumping protocol by which the system is initiated in the vacuum state of $H(\tau_1)$ and evolves from $\tau_1$ to $\tau_2$ under $H(\tau)$ is ideal and the pumped charge per cycle is equal to $-n$.

In Fig. (5) different paths on x-y plane for the x-independent solution of Sh-G equation are demonstrated with their respective pumped charge for the Rice-Mele model.

## 6 Discussion-Conclusions

As it was shown, we have provided explicit examples of perfect topological pumps that operate at finite frequency. We managed to do that by creating a map from the zero curvature repre-

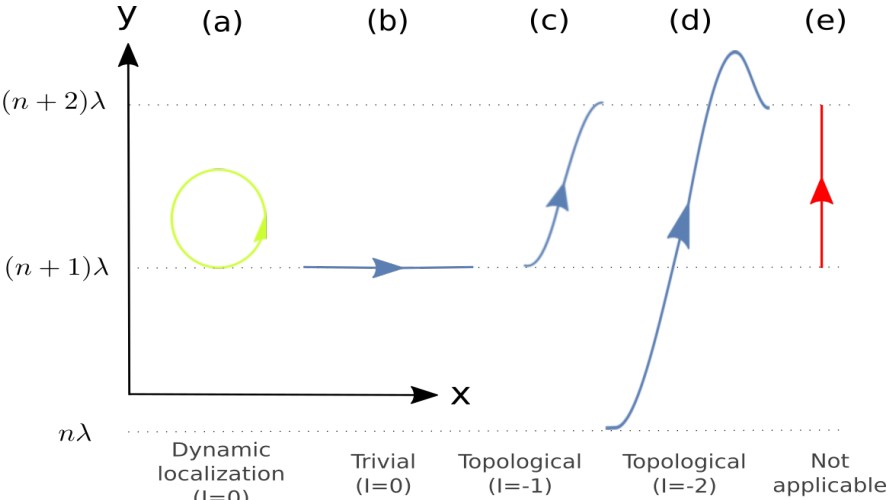

Figure 5: Different paths on x-y plane of the zero curvature of the Euclidean x-independent ShG that represent different single cycles of Rice-Mele model and the respective pumped charge $I$. The path (a) corresponds to the dynamic localization (See footnote 2 in a previous page). The path (b) is referred as Trivial because the whole evolution is performed by $i\hat{A}_x$ (See Discussion for more). The paths (c),(d) are the main result of our work, where an exact integer of charge is pumped. Last, the path (e) is not compatible with Theorem 1.

sentation of ShG equation to the Rice-Mele model. A first comment is this map is not bijective, i.e. every adiabatic configuration of R-M model is not possible to be mapped back to the ShG. In fact, the examples we propose are considerably different from the ones usually used in the study of Rice-Mele model; the relations (23) suggest complex hopping parameters, i.e. with presence of a homogeneous electric field.

Moreover, many parts of our proofs/derivations were model-independent, i.e. did not use the particular form of the matrices. The only real restriction is a proper correspondence between the quasi-momentum and a free parameter; typically the solution of the integrable equations through that involves the introduction or auxiliary parameters, the so-called spectral parameters. This is what we exploited to construct the mapping. After all, there is open space by using the same train of thought on different integrable PDEs that are already known. This may create even wider families of perfect fast Thouless pumps.

A crucial part for the validity of the proposed Theorem 1 is that the initial configuration to be $h_\gamma(0,k) \propto i\hat{A}_x, \forall k$. This is important, since Eq. (16) for the x-independent case yields:

$$\partial_y \hat{A}_x + [\hat{A}_x, \hat{A}_y] = 0 \quad \Rightarrow \quad \partial_\tau \hat{A}_x - i[\hat{A}_x, \hat{h}_\gamma(\tau)] = 0. \tag{31}$$

This means the $\hat{A}_x$ (or $i\hat{A}_x$, if we want it Hermitean) is the dynamical (Lewis-Riessenfeld) invariant [44] for the Hamiltonian $\hat{h}_\gamma$ defined in Eq. (20). It is well-known [45] that the eigenvectors of the invariant coincide with the Floquet basis; thus the condition for noiseless cycles of the fast pump is ensured. Nevertheless, the interesting part is that finding the dynamical invariants is a formidable task, in general. But with our particular example, both identifying it and preparing the system to its lowest energy Floquet basis are straight-forward. Also a novel addition is the exact pumped charge per period at any finite frequency. Although, in general, there has been argued [11] that by preparing the system in the lowest energy Floquet state, the divergence from perfect integer is exponentially smaller than preparing it in some eigenstate of the Hamiltonian, still we have proven our example retains perfect integer pumped charge at the end of the cycle for any finite-frequency. This exceptional behaviour is

probably due to the integrabality of the ShG we used to perform the mapping to Rice-Mele model.

Finally, we would like to comment on the implementation of the proposed protocol with ultracold Fermions in optical lattices [42, 46]. A first study about the stability was performed in [47] and has investigated the effects of factors that are present in a realistic setting beyond the fine-tuned setting we presented. In particular, it showed that the corrections are of first order with the next-to-nearest neighbour interaction term and of second order in the parabolic confining potential. Nevertheless, these are not a huge obstacles for a realistic implementation, as the ultracold atomic experiments have the advantage of high control of these parameters. Also it made clear that the finite-size effects can easily be suppressed and to have perfect quantization with very small number of fermions. Moreover, it is worth noting that atoms in an optical lattice are neutral, therefore the "electric field" pulse (11) has to be simulated by accelerated motion of the lattice potential as a whole. At present we do not understand the effects of interparticle collisions on the fidelity of the protocol, however it should be borne in mind that such collisions are rather inefficient in spin-polarized fermionic systems.

# Acknowledgements

This publication is part of the project Adiabatic Protocols in Extended Quantum Systems, Project No 680-91-130, which is funded by the Dutch Research Council (NWO).

# A    Appendix

## A.1    Proof of Lemma 1

For convenience, we assume $x_1 = 0$. Generalisation to other starting points is trivial. It follows directly from Eqs. (20) and (29) that

$$\hat{h}_{\gamma^a}(\tau) = \left\{ i\alpha^{-1}\hat{A}_x(\mathbf{x}, k) + in\lambda\hat{A}_y(\mathbf{x}, k) \right\}\big|_{\mathbf{x}=\gamma^a(\tau)},$$

where both $\hat{A}_y(\gamma^a(\tau), k)$ and $\hat{A}_y(\gamma^a(\tau), k)$ are smooth (of class $\mathbb{C}^\infty$) bounded $\alpha$ - independent functions of $\tau$ and $k$. Furthermore,

$$i\hat{A}_x(\gamma^a(\tau), k) = \hat{\sigma}_x \delta(\tau)\sin\frac{k}{2} + \hat{\sigma}_y t(\tau)\cos\frac{k}{2} + \hat{\sigma}_z m(\tau), \tag{32}$$

where the functions $\delta(\tau)$ and $m(\tau)$ are defined in Eq. (28) and

$$t(\tau) = -\frac{1}{2}\cosh\left[\frac{\varphi(n\lambda\tau)}{2}\right].$$

Denote by $\epsilon_+(\tau, k)$ the non-negative eigenvalue of $\hat{h}_{\gamma^a}(\tau)$. It is easily seen that the second eigenvalue is given by $\epsilon_-(\tau, k) = -\epsilon_+(\tau, k)$ Since, according to Proposition 1, the path $p(\tau) = (\delta, m)(\tau)$ avoids the origin $(0, 0)$ and $|t(\tau)| > 0$, for all $\tau$, the spectrum of the matrix (32) has a gap for all $k \in [0, 2\pi)$ and for all $\tau \in [0, 1]$. Therefore, in a small enough vicinity of $\alpha = 0$, there exists $\Delta > 0$ such that

$$\epsilon_+(\tau, k) - \epsilon_-(\tau, k) > \Delta, \tag{33}$$

for all $\alpha$, $k$ and $\tau$. We note that $\epsilon_+^\alpha(\tau, k)$ is a smooth bounded function of $\tau$ and $k$, which is also $2\pi$-periodic in $k$.

We now use the asymptotic estimate by Kato [48]

$$\hat{U}_{\gamma^\alpha}(k)\hat{P}^\alpha_\pm(0,k) = e^{\mp i\alpha^{-1}\vartheta^\alpha(k)}\hat{W}^\alpha_\pm(k)\hat{P}^\alpha_\pm(0,k) + O(\alpha), \tag{34}$$

where $\hat{P}^\alpha_{+(-)}(\tau,k)$ is the projection matrix onto the positive (negative) eigenspace of $\hat{h}_{\gamma^\alpha}(\tau)$,

$$\vartheta^\alpha(k) = \int_0^1 \epsilon^\alpha_+(\tau,k)d\tau, \tag{35}$$

and $\hat{W}^\alpha_\pm$ is related to the unitary evolution matrix $\hat{V}^\alpha_\pm(\tau)$ satisfying:

$$\frac{d}{dt}\hat{V}^\alpha_\pm(\tau,k) = \left[\frac{d}{dt}\hat{P}^\alpha_\pm(\tau,k), \hat{P}^\alpha_\pm(\tau,k)\right]\hat{V}^\alpha_\pm(\tau,k),$$
$$\hat{V}^\alpha(0,k) = \mathbb{I}, \tag{36}$$

by $\hat{W}^\alpha_\pm(k) = \hat{V}^\alpha_\pm(1,k)\hat{P}^\alpha_\pm(0,k)$. As is shown in [48] $\hat{W}_\pm(k)\hat{P}^\alpha_\pm(0,k) = \hat{P}^\alpha_\pm(1,k)\hat{W}_\pm(k)$ and $[\hat{W}^\alpha_\pm(k)]^\dagger\hat{W}^\alpha_\pm(k) = \hat{P}^\alpha_\pm(0,k)$. Since $\hat{P}^\alpha_\pm(0,k) = \hat{P}^\alpha_\pm(1,k)$ we conclude that

$$\hat{W}^\alpha_\pm(k) = \hat{P}^\alpha_\pm(0,k)e^{i\beta^\alpha_\pm(k)}, \tag{37}$$

where $\beta^\alpha_\pm(k)$ are the topological Berry phases [4,49]. We note that the projectors $\hat{P}^\alpha_\pm$ together with their derivatives are smooth $2\pi$-periodic in $k$ bounded functions of $k$ and $\tau$ in any vicinity of $\alpha = 0$ therefore $\hat{W}^\alpha_\pm(k) = \hat{W}^0_\pm(k) + o(1)$, $\alpha \to 0$ where

$$\hat{W}^0_\pm(k) = P^0_\pm(0,k)e^{i\beta^0_\pm(k)}$$

are smooth $2\pi$-periodic functions of $k$. The Kato estimate (34) and equation (37) imply that

$$\hat{U}_{\gamma^\alpha}(k) = \hat{P}^\alpha_+(0,k)e^{-i\alpha^{-1}\vartheta^\alpha(k)+i\beta^\alpha_+(k)} + \hat{P}^\alpha_-(0,k)e^{i\alpha^{-1}\vartheta^\alpha(k)+i\beta^\alpha_-(k)} + O(\alpha), \quad \alpha \to 0. \tag{38}$$

The existence of a lower bound (33) on the spectral gap of the operator $\hat{h}_{\gamma^\alpha}(\tau)$ in a small vicinity of $\alpha = 0$ implies that the eigenvalues $\epsilon^\alpha(k)$ and the projectors $P^\alpha_\pm(0,k)$ admit for uniform perturbative estimates [50]

$$\epsilon^\alpha(k) = \epsilon^0(k) + \alpha\tilde{\epsilon}(k) + O(\alpha),$$
$$\hat{P}^\alpha_\pm(0,k) = \hat{P}^0_\pm(0,k) + O(\alpha), \quad \alpha \to 0. \tag{39}$$

This, in particular, implies

$$\vartheta^\alpha(k) = \vartheta^0(k) + \alpha\tilde{\vartheta}(k) + O(\alpha), \quad \alpha \to 0, \tag{40}$$

where $\vartheta^0(k)$ and $\tilde{\vartheta}(k)$ are smooth $2\pi$- periodic functions of $k$ given by

$$\vartheta^0(k) = \int_0^1 d\tau\varepsilon^0(\tau,k), \qquad \tilde{\vartheta}(k) = \int_0^1 d\tau\tilde{\varepsilon}(\tau,k).$$

Using estimates (39) and (40) in (38) we find

$$\hat{U}_{\gamma^\alpha}(k) = \hat{G}_+(k)e^{-i\alpha^{-1}\vartheta^0(k)-i\tilde{\vartheta}(k)+i\beta^0_+(k)} + \hat{G}_-(k)e^{i\alpha^{-1}\vartheta^0(k)+i\tilde{\vartheta}(k)+i\beta^0_-(k)} + O(\alpha), \quad \alpha \to 0, \tag{41}$$

where $\hat{G}_{+(-)}(k)$ is the projector onto the positive (negative) eigenspace of the matrix

$$\hat{g}(k) = i\hat{A}_x(0,k) = \hat{G}_+(k)\epsilon^0_+(0,k) - \hat{G}_-(k)\epsilon^0_+(0,k). \tag{42}$$

We now define the function

$$\chi(k) = \frac{\vartheta_0(k)}{\epsilon_+^0(0,k)}.$$ (43)

Then there exists the limit

$$\hat{\omega}(k) = \lim_{\alpha \to 0} e^{i\alpha^{-1}\chi(k)\hat{g}(k)} \hat{U}_{\gamma^\alpha}(k) = \hat{G}_+(k) e^{-i\tilde{\vartheta}(k)+i\beta_+^0(k)} + \hat{G}_-(k) e^{i\tilde{\vartheta}(k)+i\beta_-^0(k)},$$ (44)

which establishes the statement of item (i) of the Lemma.

We now proceed to evaluating the integral

$$I\left[\hat{\omega}, \hat{G}_-\right] = -i \oint \frac{dk}{2\pi} \text{Tr}[\hat{G}_-(k)\hat{\omega}^{-1}(k)\partial_k \hat{\omega}(k)],$$ (45)

where the integration is performed over the period $2\pi$ of $\hat{G}_-(k)$ and $\hat{\omega}(k)$. Substituting Eq. (44) into (45) and taking into account the periodicity of $\tilde{\vartheta}(k)$ we have

$$I[\hat{\omega}, \hat{G}_-] = \oint \frac{dk}{2\pi} \partial_k \beta_-^0(k).$$ (46)

The $2\pi$-periodicity of $\hat{W}_-^0(k)$ implies that this integral is an integer. The value of this integer is not affected by the replacement $\hat{\omega}(k) \mapsto \hat{\omega}(k)^{ia\hat{g}(k)+b\chi(k)\hat{g}(k)}\hat{\omega}(k)$ because both $\epsilon_+(k)$ and $\chi(k)$ are smooth $2\pi$-periodic functions of $k$. Recalling that $\beta_-^0(k)$ has the meaning of the Berry phase induced by the adiabatic evolution under the Hamiltonian (32) we use the well known result, see e.g. Ref. [6], that the integral (46) coincides with the winding index of the curve $(\delta, m)(\tau)$ around the point $(0,0)$. By virtue of Proposition 1, this number is given by $-n$, which amounts to property (ii).

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
