# Peer review of "An ideal rapid-cycle Thouless pump"

_SciPost Physics, doi:SciPost Phys. 12, 203 (2022)_

## Round 2 · Referee Report · Anonymous (Referee 1) · 2021-12-13

Report

The paper shows an interesting link between the Rice-Mele model and the Euclidean Sinh-Gordon equation. The authors use this to device a driving protocol which realizes exact charge pumping across a one-dimensional chain. The effect is similar to Thouless’ charge pump. However, whereas Thouless pump only realizes quantized pumping in the adiabatic limit, the current proposal realizes exact pumping at finite modulation speed. This is a nice result, which might be suited for publication in SciPost, as the protocol for exact charge pumping provides a significant theoretical result. (C.f. the first of SciPost Physics’ acceptance expectations).

Unfortunately, the paper is very hard to read. Even though the results appear to be solid and interesting, it is my judgement that the manuscript is too inaccessible in its current form to be useful for the community. In particular, it does not meet the first general acceptance criteria of SciPost, namely that it “must be written in a clear and intelligible way, free of unnecessary jargon, ambiguities and misrepresentations.” Hence I cannot recommend publication. This is unfortunate, since I believe the results are significant and interesting. I would thus strongly recommend the authors to improve the clarity and level of explanation in the paper, as the difficulty of reading the paper unnecessarily limits its appeal and impact.

The paper is written in a mathematical style (using “proof”, “Lemma”, etc.). However, in some instances it lacks the mathematical rigor and clarity which is required for this style. For instance, theorem 1 reads

“Let $\phi(\mathbf{x}) = \varphi(y)$ and let $\lambda$ be it fundamental period. Let $\gamma \in \Gamma_{n\lambda}$, where $n\in {\mathbb Z}$ then (i) $\gamma$ is orderly, and (ii) $\nu_\gamma = -n$”.

There is nothing in this statement that links $\gamma$ to $\phi$ or $\varphi$, except through the existence of the scalar $\lambda$. Moreover, as the quantities are defined, the relationship between $\varphi$ and $\phi$ does not play any role for the properties of $\gamma$. As I understand the properties (i-ii) of $\gamma$ rely on other properties of $\phi$ than the existence of $\lambda$.

Another example is Eq. (2) where there is nothing on the right-hand side which depends on $\tau_1$ or $\tau_2$.

The mathematical styling of the paper has the disadvantage that the physics, meaning of statements and the motivation of steps are harder to decode. This is usually compensated by the fact that statements are clear and rigorous. However, this advantage is not present in the current paper. As a result, the paper is very hard to read. In any case, since the manuscript is a physics paper, I believe a more “physics”-like style of the discussion would give it a broader appeal.

Apart from the mathematical style, the general discussion also appears unnecessarily complicated at many places: even though I am very familiar with the field I even found it challenging to read the introduction and Sec. II, which review the basic features of the Rice-Mele model and charge pumping. I am convinced these aspects can be explained in a much simpler way that non-specialists would understand.

A good example is Eq. (2): Eq. (2) in essence states that the time-evolution can be decomposed as a direct product of evolution operators at specific crystal momenta. However, Eq. (2) states in a very indirect way which makes use of a lot of new, unfamiliar, and apparently arbitrary notation: for example, why does the many-body state $|\Psi\rangle$ appear as a ket, but not the spinor states $v(k)$? And why does $\tau_1$ and $\tau_2$ not enter on the right hand side? It would be much easier to understand the statement in Eq. (2) if the authors also provided an explanation of this result with words

The same style/level of explanation, which persists throughout the paper, makes it challenging for me as a specialist to even decode the results I am familiar with. Because of the same style, it is even harder for me (and other readers) to understand the new results which are obtained in the paper.

I provide specific comments below:

1) the notation and formalism is unnecessarily non-standard. The nonstandard terms “orderly” and “ideal” cycles play a prominent role, along with “natural” and “Floquet-proper”. I find it hard to see the motivation for this terminology, and the unfamiliar language makes the paper hard to read. The designation of a cycle as “natural” and “Floquet-proper“ means that the eigenstates of the Floquet operator coincides with the eigenstates of the initial Hamiltonian. Such a direct description would me much easier to understand.

2) Related to the above, the paper would benefit a lot from a more extensive discussion of the physics of the results. For instance it appears that the electric field “E” is crucial for ensuring the exact quantization of charge pumping (c.f. Fig. 4). A discussion of this fact would be illuminating.

3) There are several examples of typos, some of which appear in crucial statements of the paper (such as theorems) . This unnecessarily complicates the reading of the text. For instance, Theorem 1, which is quoted above, appears to lack a period, which makes the theorem hard to parse. As another example, footnote 1 lacks a period. Below Eq. (2), there is no hat on the symbol $U$, which suggests that this is no the same symbol which enters in Eq. (2). Finally, above Eq. 22: “(…) are determined by the propereties of the path $\gamma$ as follows $v(\tau)=(…)$”. Here a colon appears to be missing.

The existence of these (and other) typos moreover suggests the paper have not been properly worked through before submission. I therefore suggest the authors give the paper a thorough readthrough before the next submission.

4) On the physics side, the exact charge pumping described requires fine-tuning. The advantage of the Thouless pump is that is topologically robust; i.e., that quantization of charge pumping is not affected by perturbations as long as the driving cycle remains adiabatic. This therefore limits the scope and applicability of the findings relative to the Thouless pump. Comments on this would be very helpful.

5) Fig. 4 (c) compares the “exact” protocol devised by the author (solid line) with the “standard” Thouless charge pump (dashed line). However, the protocol used for the latter data does not appear to be in the adiabatic regime, and the data includes the first few periods of the cycle. Would it be possible to show similar data with a Thouless pump in the adiabatic regime, and over more cycles of the protocol?

  • validity: good
  • significance: good
  • originality: high
  • clarity: poor
  • formatting: mediocre
  • grammar: below threshold

Author:  Savvas Malikis  on 2022-05-03  [id 2436]

(in reply to Report 1 on 2021-12-13)

First of all, we want to thank the referee for their report. Their comments helped us improve the manuscript and put it in perspective.

In the newer version, the main modification we did to the manuscript is the addition of a section at the beginning where we describe the main result without technical details. This way, we humbly believe, we have managed to make our main result accessible to people who may not be interested in rigorous, however lengthy derivations. Moreover, we gave more space to the explanation the physical intuition behind the presented result. In particular, our construction relies on the zero curvature representation allowing one to relate different paths that have the same starting/ending points. Physically, the way we initialize the protocol, we prepare the system at a Floquet state (that coincides with the ground state). Lastly, we lightened some of the jargon/formulation we used in the previous version.

The referee writes:

There is nothing in this statement that links $\gamma$ to $\phi$ or $\varphi$, except through the existence of the scalar $\lambda$. Moreover, as the quantities are defined, the relationship between $\phi$ and $\varphi$ does not play any role for the properties of $\gamma$. As I understand the properties (i-ii) of $\gamma$ rely on other properties of $\phi$ than the existence of $\lambda$.

Our response: In this comment the Referee demonstrates a perfectly good understanding of the statement of the theorem. If anything, this demonstrates that the theorem is stated both clearly and rigorously enough.

The referee writes:

why does the many-body state appear as a ket, but not the spinor states?

Our response: We have changed the notation to make it more uniform. The Dirac notation is now used for all quantum states.

The referee writes:

And why does $\tau_1$ and $\tau_2$ not enter on the right-hand side? It would be much easier to understand the statement in Eq. (2) if the authors also provided an explanation of this result with words

Our response: The referee is correct in regards to the old Eq. 2. We omitted the $\tau_1,\tau_2$ for brevity. We have made the time arguments explicit in the revised version.

The referee writes:

The nonstandard terms “orderly” and “ideal” cycles play a prominent role, along with “natural” and “Floquet-proper”. I find it hard to see the motivation for this terminology, and the unfamiliar language makes the paper hard to read.

Our response: Regarding the styling/jargon we have put some effort into making it lighter. We removed the terms "natural", and "Floquet-proper" terms. We kept "ideal", which is easy to explain, because we believe it is central to explaining the importance of our result. Lastly, the term "orderly" is found in the sections with the proofs, meaning that someone has to face this term only if they are willing to follow the details of the paper. This term is completely defined in the text, it is convenient for the purposes of our discussion and it lacks any standard alternative because it refers to a specific object emerging in the context of our analysis.

The referee writes:

There are several examples of typos..

Our response: We improved the punctuation all along the text.

The referee writes:

On the physics side, the exact charge pumping described requires fine-tuning. The advantage of the Thouless pump is that is topologically robust; i.e., that quantization of charge pumping is not affected by perturbations as long as the driving cycle remains adiabatic. This therefore limits the scope and applicability of the findings relative to the Thouless pump. Comments on this would be very helpful.

Our response: We agree that this point may not be self-evident therefore we have added clarifying discussion to the manuscript. In particular, in the Section 2 we explain the problem of non-adiabatic ideal pumps and that one should restrict themselves in fine-tuned settings.

The referee writes:

However, the protocol used for the latter data does not appear to be in the adiabatic regime, and the data includes the first few periods of the cycle. Would it be possible to show similar data with a Thouless pump in the adiabatic regime, and over more cycles of the protocol?

Our response: It is in principle possible to generate plots similar to in Fig. 4 for an adiabatic cycle. However, we honestly do not see the point. The properties of an adiabatic cycle were established long ago and are recapped in our manuscript: the adiabatic fidelity is equal to 1 throughout the cycle and the pumped charge is integer. As for adding more periods to the plots, in our opinion that is unnecessary too. This is because the ideal pumping protocol is strictly periodic, as is rigorously proven in our manuscript, so adding more periods will not have any added value. In contrast, the non-ideal protocol, which is obtained from the ideal one by removing the time-dependent vector potential is added in order to illustrate the departure of the pumped charge from perfectly quantised values. Such a departure is already seen in the first few periods, therefore adding more periods is not going to change anything except making the plot busier and more difficult to read.

---

## Round 2 · Referee Report · Johann Kroha (Referee 2) · 2021-12-19

Strengths

  1. Importance for the field of topological transport
  2. Exact mathematical proof

Weaknesses

  1. No physical intuition/insight given for the results
  2. Figures 1 and 3 do not provide significant additional insight into the material discussed in the paper. Figs. 1 and 3 are not refered to in the text.

Report

In an adiabatic Thouless pump the transported charge per cycle is quantized because the system remains in its momentary ground state at any time during the cycle. However, when the driving is done non-adiabatically, i.e., at a finite driving period, the quantization is in general destroyed because excited states are created by the non-zero frequencies of the drive. Therefore, It is important to design the driving protocol in such a way that all the excitations created during a cycle are brought back to the ground state by the end of the cycle and that no overall entropy is created during the cycle. This question is closely related to the quantum brachistochrone problem, the problem of how a quantum state can be transported from one location in space to another during aminimal time without altering the state.
In the present work, the authors prove mathematically how such a non-adiabatic yet quantizing cycle protocol may be designed. One can follow this proof step by step. However, no physical insight is given how this protocol avoids excited states, or reduces the system to the ground state and avoids entropy production at the end of a cycle. Such insight would greatly improve the impact of the paper and possible enable other researchers to realize quantized driving protocols physically. Therefore, I would like to ask the authors, can they to explain or at least visualize their interesting results in terms of avoided entropy production and how the necessarily created excited states during a cycle are brought back to the system’s ground state at the end of the cycle? In particular, in what respect is the mapping between the sinh-Gordon model and the RM model crucial, or can one imagine other mappings?
As less important points, the presentation is less than optimal. Fig. 1 contains no information regarding the present topic, as it illustrates any mapping between any two parameter spaces. The readability of Fig. 3 would be improved if it would show the space would be shown in which the trajectories are embedded. I believe, that is the x axis as the horizontal axis for each of the trajectories.

The paper is correct, as far as can be checked, and important for its field of research. This warrants eventual publication in SciPost. However, the changes mentioned above are essential.

Requested changes

As described in the report.

  • validity: high
  • significance: high
  • originality: good
  • clarity: low
  • formatting: good
  • grammar: excellent

Author:  Savvas Malikis  on 2022-05-03  [id 2437]

(in reply to Report 2 by Johann Kroha on 2021-12-19)

First of all, we want to thank the referee for their report. Their comments helped us improve the manuscript and put it in perspective.

In the newer version, the main modification we did to the manuscript is the addition of a section at the beginning where we describe the main result without technical details. This way, we humbly believe, we have managed to make our main result accessible to people who may not be interested in rigorous, however lengthy derivations. Moreover, we gave more space to the explanation the physical intuition behind the presented result. In particular, our construction relies on the zero curvature representation allowing one to relate different paths that have the same starting/ending points. Physically, the way we initialize the protocol, we prepare the system at a Floquet state (that coincides with the ground state). Lastly, we lightened some of the jargon/formulation we used in the previous version. Specific comments follow addressing the particular remarks of each reviewer.

The referee writes:

However, no physical insight is given how this protocol avoids excited states, or reduces the system to the ground state and avoids entropy production at the end of a cycle. Such insight would greatly improve the impact of the paper and possible enable other researchers to realize quantized driving protocols physically

Our response: We made more clear in the new version the physical intuition why this protocol works. Essentially we construct the protocol in such a way as to make sure that the ground state of the many-body system at the initial moment of time also coincides with the Floquet eigenstate.

The referee writes:

In particular, in what respect is the mapping between the sinh-Gordon model and the RM model crucial, or can one imagine other mappings?

Our response: The crucial element of the whole construction is a mapping between a tight-binding model (the R-M model in the present case) and the zero-curvature representation of an integrable field theory. Additionally, the mapping has to use the L-M pair which is a periodic function of the real spectral parameter and in which both L and M are anti-hermitian. We know examples of other tight-binding models which can be mapped onto an integrable field theory, however, we reserve the analysis of such examples as well as the classification of such mappings for a separate study.

The referee writes:

Therefore, I would like to ask the authors, can they to explain or at least visualize their interesting results in terms of avoided entropy production and how the necessarily created excited states during a cycle are brought back to the system’s ground state at the end of the cycle?

Our response: Regarding the entropy production. Throughout the Thouless cycle the system remains in a pure state therefore the only honest way of analysing the entropy production that we know of is by computing the entanglement entropy density. However, such a computation is quite difficult and it does not add enough value to the presented result to justify its complexity.

The referee writes:

As less important points, the presentation is less than optimal. Fig. 1 contains no information regarding the present topic, as it illustrates any mapping between any two-parameter spaces. The readability of Fig. 3 would be improved if it would show the space would be shown in which the trajectories are embedded. I believe, that is the x axis as the horizontal axis for each of the trajectories.

Our response: We refer the mentioned figures in the text showing their role in the whole body of the manuscript. We added the x-axis in the old Fig. 3 to make it clear.

---

## Round 2 · Referee Report · Anonymous (Referee 3) · 2022-1-2

Strengths

  1. Important contribution to topological quantum transport
  2. Solid derivation

Weaknesses

  1. Results and their implications are difficult to discern.
  2. No physical interpretation.

Report

The Thouless pump is a theoretical paradigm for topologically protected quantization of transport. The work presented in the manuscript extend the notion of optimal pumps typically reserved to the case of adiabatic pumping, to the case of finite-frequency driving.

This topic is of significant theoretical importance, and the results of the paper seem very solid. However, the results and their implications seem to target a readership with a mathematical physics orientation and are not very accessible to the condensed matter community. This substantially hinders the impact of the work. Therefore, in its present form, the manuscript does not meet the general acceptance criteria of publication in SciPost: 1) Be written in a clear and intelligible way, free of unnecessary jargon, ambiguities and misrepresentations. and 6) Contain a clear conclusion summarizing the results (with objective statements on their reach and limitations) and offering perspectives for future work.

In order to increase the impact of the manuscript, I strongly recommend that the authors express the results, their interpretation and significance in more physical terms. In particular, I would like the authors to address the following questions:

1) What do these optimal protocols correspond to physically? 2) What is the feature that makes these particular protocols optimal? Is there a typical time scale that protects the pumping? An effective symmetry that constrains certain transitions? Do they correspond to a transformation onto an effective adiabatic cycle? 3) Are these protocols unique? Is the quantized transport robust to small changes in the driving protocol? 4) How do these results relate to previous works on quantized pumping in Floquet driven systems? Such as Int. J. Mod. Phys. B vol.18, 3071 (2004), Phys. Rev. B 82, 235114 (2010), Phys. Rev. Lett. 120, 150601 (2018) to name a few. 5) While the authors discuss a particular example in the Rice Mele model - the discussion is rather abstract and the physical interpretation of the parameters and their respective modulations remains unclear. 6) What if any are the implications of these optimal protocols on more “realistic” driving protocols? Is it possible to quantify deviations from quantization based on perturbations around these optimal protocols? Can we use these results to identify nearly quantized transport in highly non adiabatic cases? 7) If their physical interpretation is unclear, what is the significance of these protocols on the field of quantum transport?

  • validity: high
  • significance: high
  • originality: high
  • clarity: poor
  • formatting: reasonable
  • grammar: reasonable

Author:  Savvas Malikis  on 2022-05-03  [id 2438]

(in reply to Report 3 on 2022-01-02)

First of all, we want to thank the referee for their report. Their comments helped us improve the manuscript and put it in perspective.

In the newer version, the main modification we did to the manuscript is the addition of a section at the beginning where we describe the main result without technical details. This way, we humbly believe, we have managed to make our main result accessible to people who may not be interested in rigorous, however lengthy derivations. Moreover, we gave more space to the explanation the physical intuition behind the presented result. In particular, our construction relies on the zero curvature representation allowing one to relate different paths that have the same starting/ending points. Physically, the way we initialize the protocol, we prepare the system at a Floquet state (that coincides with the ground state). Lastly, we lightened some of the jargon/formulation we used in the previous version. Specific comments follow addressing the particular remarks of each reviewer.

The referee writes:

What do these optimal protocols correspond to physically?

Our response: We have serious difficulty addressing this point because the Referee's question is too vague. The protocols are literally the instruction as to how the tight-binding parameters of a physical system described by the Rice-Mele Hamiltonian need to be changed as a function of time in order to achieve noise-free topologically quantised pumping of charge at finite frequency. How more physical can it get?

The referee writes:

What is the feature that makes these particular protocols optimal? Is there a typical time scale that protects the pumping? An effective symmetry that constrains certain transitions? Do they correspond to a transformation onto an effective adiabatic cycle?

Our response: The maximum frequency the pump can work, as we explain in the text, is close to the gap. The reason is if we try to speed up the protocol further, this will lead to increase of the gap. The only symmetry we have is the translational symmetry, i.e. the Fourier transform that allows only transmissions between the negative and positive eigenstates of every quasi-momentum. Last, as we explain in the paper, the protocols have an adiabaticity parameter $\alpha$ and when $\alpha\rightarrow 0$, this corresponds to an adiabatic protocol.

The referee writes:

Are these protocols unique? Is the quantized transport robust to small changes in the driving protocol?

Our response: The protocols are not unique in the sense that we performed a particular mapping. It is fine-tuned, but there is freedom. Moreover, we performed the mapping for a particular example of an integrable equation. So it's possible that one can do the same with other equations, creating other protocols.

The referee writes:

How do these results relate to previous works on quantized pumping in Floquet-driven systems? Such as Int. J. Mod. Phys. B vol.18, 3071 (2004), Phys. Rev. B 82, 235114 (2010), Phys. Rev. Lett. 120, 150601 (2018) to name a few.

Our response: The second reference is strongly related to our work; essentially this is the topological invariant we use to compute the pumped charge. But the other two references are not related, since they have a completely different approach.

The referee writes:

What if any are the implications of these optimal protocols on more “realistic” driving protocols? Is it possible to quantify deviations from quantization based on perturbations around these optimal protocols? Can we use these results to identify nearly quantized transport in highly non-adiabatic cases?

Our response: Our paper presents an example of an "integrable" or "exactly solvable" class of protocols (in the same sense as Onsager's solution of the Ising model presents an integrable example of a second-order phase transition). As with all integrable models, the analysis of deformations around the exactly solvable point is a huge topic. The first stab at such analysis is made in 10.1103/PhysRevA.104.063315. But there is plenty more to explore, for example realizing the protocol in an optical lattice. This is definitely a matter for (a lot of) further work...

The referee writes:

If their physical interpretation is unclear, what is the significance of these protocols on the field of quantum transport?

Our response: The notion of ''physical interpretation`` is subjective and has no codified meaning therefore we cannot address this point. As for the significance of the protocols, they represent the first to our knowledge example of topologically quantized noise-free pumping at finite frequency, that is without the protection of adiabaticity.

---

## Round 3 · Referee Report · Anonymous (Referee 3) · 2022-5-17

Strengths
Solid derivation and mathematical proofs
Well written introduction which summarizes the results and clarifies their limitations
Well written introduction which summarizes the results and clarifies their limitations
Weaknesses
Not generalizable. limited impact
Report
I would like to thank the authors for their detailed response. I found the new introduction and overview sections very helpful and very well written.
It is now clarified that the finite frequency protocol suggested in the manuscript is in fact fine tuned and “engineered” such as to lead to quantized pumped charge without noise generation. Consequently, any deviations from the fine tuned limit would most probably result in noise generation and loss of quantization.
As a result of the protocols being fine tuned, I find the implications of the work on the field of topological quantum pumping somewhat limited since they cannot be generalized to more realistic scenarios, and are most likely not immediately relevant for experiments. Moreover, the manuscript does not allude to any physical intuition as to why the chosen protocols avoid excitations (although the analogy of to the quantum brachistochrone seems intriguing).
In conclusion, I believe that the topic is very interesting, the work is solid and while the immediate implications of the results are limited, they offer perspective for followup work on the topic of protected finite frequency pumping. I therefore recommend the manuscript for publication.
It is now clarified that the finite frequency protocol suggested in the manuscript is in fact fine tuned and “engineered” such as to lead to quantized pumped charge without noise generation. Consequently, any deviations from the fine tuned limit would most probably result in noise generation and loss of quantization.
As a result of the protocols being fine tuned, I find the implications of the work on the field of topological quantum pumping somewhat limited since they cannot be generalized to more realistic scenarios, and are most likely not immediately relevant for experiments. Moreover, the manuscript does not allude to any physical intuition as to why the chosen protocols avoid excitations (although the analogy of to the quantum brachistochrone seems intriguing).
In conclusion, I believe that the topic is very interesting, the work is solid and while the immediate implications of the results are limited, they offer perspective for followup work on the topic of protected finite frequency pumping. I therefore recommend the manuscript for publication.

Savvas Malikis on 2022-05-16 [id 2470]
A couple of typos have come to our attention in the new version.
In particular, in Eq. 8 it is $\dot v(\tau)$ (instead of $v(\tau)$).
Also in the second half of Eq. 12 for the $t_{1,2}$ term, on the numerator it is $V(\tau)$( instead of $v(\tau)$).

---

## Round 3 · Referee Report · Anonymous (Referee 1) · 2022-5-19

Strengths
Exact results
Well written introduction
Well written introduction
Weaknesses
Fine-tuning required for protocol means the result have limited physical impact
Report
The clarity of the manuscript is significantly improved; the new version of the manuscript is clearly structured and well-written. The addition of the new Sec II makes the physics of the result much more clear, without sacrificing the rigor.
While the physical impact of the results is limited due to the fine-tuning required, the simple form of the exact charge-pumping protocol may be of use for further theoretical studies.
I have no more reservations recommend the manuscript for publication.
While the physical impact of the results is limited due to the fine-tuning required, the simple form of the exact charge-pumping protocol may be of use for further theoretical studies.
I have no more reservations recommend the manuscript for publication.

---

## Round 3 · List of Changes

In this new version, we added Section 2 explaining the take-home message of our paper.
Reading the reports, we realized that the previous version was hard to read, since it had the structure of a mathematical physics paper, even though it presents a physically remarkable result. Thus to make it accessible, without sacrificing the rigorous derivations, we added that part.
Reading the reports, we realized that the previous version was hard to read, since it had the structure of a mathematical physics paper, even though it presents a physically remarkable result. Thus to make it accessible, without sacrificing the rigorous derivations, we added that part.

---

## Editorial Decision

published